# The Hen or the Egg: Impaired Alveolar Oxygen Diffusion and Acute High-altitude Illness?

**DOI:** 10.3390/ijms20174105

**Published:** 2019-08-22

**Authors:** Heimo Mairbäurl, Christoph Dehnert, Franziska Macholz, Daniel Dankl, Mahdi Sareban, Marc M. Berger

**Affiliations:** 1Translational Lung Research Center (TLRC), part of the German Center for Lung Research (DZL), University of Heidelberg, 69120 Heidelberg, Germany; 2Medbase Checkup Center, 8400 Zürich, Switzerland; 3Department of Anesthesiology, Perioperative and General Critical Care Medicine, University Hospital Salzburg, 5020 Salzburg, Austria; 4University Institute of Sports Medicine, Prevention and Rehabilitation, and Research Institute of Molecular Sports Medicine and Rehabilitation, Paracelsus Medical University, 5020 Salzburg, Austria

**Keywords:** high-altitude pulmonary edema, acute mountain sickness, oxygen diffusion limitation

## Abstract

Individuals ascending rapidly to altitudes >2500 m may develop symptoms of acute mountain sickness (AMS) within a few hours of arrival and/or high-altitude pulmonary edema (HAPE), which occurs typically during the first three days after reaching altitudes above 3000–3500 m. Both diseases have distinct pathologies, but both present with a pronounced decrease in oxygen saturation of hemoglobin in arterial blood (SO_2_). This raises the question of mechanisms impairing the diffusion of oxygen (O_2_) across the alveolar wall and whether the higher degree of hypoxemia is in causal relationship with developing the respective symptoms. In an attempt to answer these questions this article will review factors affecting alveolar gas diffusion, such as alveolar ventilation, the alveolar-to-arterial O_2_-gradient, and balance between filtration of fluid into the alveolar space and its clearance, and relate them to the respective disease. The resultant analysis reveals that in both AMS and HAPE the main pathophysiologic mechanisms are activated before aggravated decrease in SO_2_ occurs, indicating that impaired alveolar epithelial function and the resultant diffusion limitation for oxygen may rather be a consequence, not the primary cause, of these altitude-related illnesses.

## 1. Introduction

Acute mountain sickness (AMS) and high-altitude pulmonary edema (HAPE) can develop within hours to days after arrival at high-altitude. Although AMS and HAPE are diseases with different pathologies, data from studies performed at the Capanna Regina Margherita at an altitude of 4559 m in the Monte Rosa mountain range, where speed and time-course of ascent were always standardized to minimize the impact of confounding factors, show that one to two days after ascent subjects suffering from AMS and/or HAPE shared a significantly higher degree of hypoxemia than individuals tolerating this altitude well. In fact, in both altitude illnesses the arterial oxygen saturation (SO_2_) was approximately 10% lower than in the healthy individuals at an altitude of 4559 m at the Capanna Margherita (AMS: e.g., [1]; HAPE: e.g., [2]), where the similarity in the magnitude of decrease is likely pure coincidence. The aggravated decrease in SO_2_ can be caused by a variety of mechanisms, including impaired ventilation, which might result in decreased alveolar PO_2_, hindrance of oxygen diffusion from the alveolar space into blood due to alveolar and interstitial edema, ventilation/perfusion mismatch, and altered oxygen affinity of hemoglobin. However, it is unclear whether the decreased SO_2_ is a prerequisite for causing high-altitude illness, or whether it is a consequence that rather aggravates severity.

Studies on the pathophysiology of AMS and HAPE have been performed on mountaineers after (rapid) ascent to altitudes >3000 m by driving, using a cable car, going on foot, and their combinations, most of which quite closely resemble mountaineering and high-altitude tourism. Studies have also been performed in chambers where in most instances exposure to normobaric hypoxia begun instantly by just stepping into a pre-adjusted hypoxic atmosphere. Both approaches have their advantages and disadvantages. Chamber studies allow for exactly-timed laboratory and clinical assessments under tightly controlled laboratory conditions. They also allow use of equipment that is not available in many high-altitude settings. By contrast, results from field studies are of course most relevant for mountaineering but are often affected by confounding factors like the physical effort of climbing a mountain, the slowly raising degree of hypoxia during the ascent, and (adverse) weather conditions. Thus it is possible that pathophysiologic mechanisms causing AMS and HAPE are different in field and laboratory conditions. Despite this possible limitation, data from these two situations will be combined in this review in order to obtain a timeline of the development of AMS and HAPE and to compare the early phase (hours, chamber studies) with the later phase (days) in field/mountain conditions. This approach might help dissect different pathophysiologic mechanisms and clarify the possible involvement of the alveolar epithelium and lung fluid balance in the pathogenesis of these diseases.

## 2. High-altitude Pulmonary Edema

HAPE typically occurs 24 to 72 h after rapid ascent to altitudes higher than 3000 m (for a comprehensive review see [3]). The incidence is approximately 5% in non-selected, not-acclimatized individuals after ascent to 4559 m [4,5]. Slow ascent rates and pre-acclimatization decrease the risk of occurrence, while the prevalence is highly increased (>60%) in individuals with HAPE during previous exposures to high-altitude indicating individual susceptibility [6].

Clinical symptoms of HAPE are dyspnea during exercise and later also at rest, cyanosis and hypoxemia, gurgling, and white, and as the illness progresses, pink, frothy sputum. X-ray images of the thorax show visible infiltrates [7]. Doppler-echocardiographic [8] and right-heart catheter [9,10] evaluations indicate exaggerated hypoxic pulmonary vasoconstriction (HPV) and increased capillary pressure [10] caused by venoconstriction or uneven distribution of blood flow [10,11]. Broncho-alveolar lavage fluid contains significant numbers of blood cells and plasma proteins but lacks typical inflammation markers [12], indicating that the alveolar leak is induced by high pressure rather than by inflammation [4]. This notion is supported by the fact that HAPE can be prevented by decreasing pulmonary arterial systolic pressure (sPAP) with, for example, nifedipine [13] and tadalafil [2]. The reasons for the exaggerated pressure response in HAPE-susceptible individuals have not been fully explored. They might include mechanisms decreasing the PO_2_ at the pulmonary micro-vasculature, such as impaired ventilation and oxygen diffusion, where the diffusion limitation might be caused by an imbalance between filtration and alveolar fluid reabsorption, but might also be based on altered sensitivity to hypoxic vasoconstriction and vasodilatory mechanisms.

### 2.1. HAPE and Ventilation

Changes in ventilation upon exposure to hypoxia follow a distinct pattern [14], showing fast stimulation followed by ventilatory decline within minutes, after which ventilation increases in the course of the next few hours and days (ventilatory acclimatization). Oxygen saturation during exposure changes accordingly. The degree of stimulation of ventilation is often used to test the sensitivity of the ventilatory drive in hypoxia. However, this test often covers only the early phase of stimulation of ventilation by hypoxia because the test procedure often lasts less than 10 min. Hohenhaus et al. [15] have shown that in the early phase of ventilatory adjustments to hypoxia, HAPE-susceptibles have a decreased isocapnic hypoxic ventilatory response (HVR) at rest, which persists during exercise. Ventilation was found to be increased by 1.5 L/min per percent decrease in SO_2_ in healthy controls but only by 0.8 L/min in HAPE-susceptible individuals. Both at rest and during exercise there was also a tendency towards lower poikilocapnic HVR in HAPE-susceptibles [15]. Thus it is possible that a blunted hypoxic ventilatory drive results in a decreased alveolar PO_2_ relative to healthy individuals. This would result in an even further decreased PO_2_ at pulmonary arterial smooth muscle cells and subsequently exaggerated hypoxic vasoconstriction, which is a hallmark of HAPE-susceptibility [13].

It is interesting to note that despite decreased HVR arterial partial pressure of carbon dioxide (PCO_2_) was not increased in HAPE-susceptibles at high-altitude (4559 m) but tended to be slightly decreased [2]. Similarly, during 2 h of exposure to normobaric hypoxia equivalent to 4559 m of altitude HAPE-susceptibles showed decreased end-tidal PCO_2_ (Figure 1). Together these results on PCO_2_ obtained in poikilocapnic conditions after 10 min to 2 h of exposure to a constant level of hypoxia indicate stimulated rather than blunted ventilation upon exposure to hypoxia for hours to days, which is in contrast to the decreased isocapnic HVR mentioned above during the HVR-test [15]. It might be that the difference in time course of reaching certain levels of hypoxia and the duration of exposure to hypoxia explains this discrepancy. A hyperventilation induced decrease in PCO_2_ might to some extent protect from exaggerated hypoxic pulmonary hypertension because low PCO_2_ acts as a vasodilator on pulmonary arteries [16], but effects of CO_2_ on HAPE-susceptibility have not been studied. Together these results show that alveolar ventilation is not abnormally low in HAPE-susceptibles during prolonged exposure to hypoxia and that a low alveolar PO_2_ is not the reason for more pronounced hypoxemia and exaggerated HPV.

### 2.2. HAPE, Alveolar Fluid Clearance, and Pulmonary Arterial Hypertension

Small amounts of fluid can easily be removed from the alveolar surface. This reabsorption is driven by an osmotic gradient generated by the reabsorption of Na^+^ across the alveolar epithelium, where Na^+^ passively enters alveolar epithelial cells via epithelial Na-channels (ENaC) inserted into the apical plasma membrane. Na^+^ is then removed actively across the basolateral membrane by the Na/K-ATPase in exchange with K^+^ [18]. Hypoxia of cultured primary alveolar epithelial cells inhibits active Na-reabsorption [19] by decreasing mRNA [20,21] and surface expression of both ENaC [22,23] and of the Na/K-ATPase [24]. Also the removal of fluid instilled into the lung was blunted when rats had been exposed to hypoxia [21,25,26], resulting in pulmonary edema indicated by increased lung water content of hypoxic rats. Measurements of alveolar reabsorption on ex-vivo human lungs have also been performed showing ENaC and Na/K-ATPase mediated active reabsorption [27], similar to rat lungs, but the effects of hypoxia have not been studied. In vivo measurements of reabsorption on the human lung are difficult. Indirect evidence can be derived from changes in the protein content of broncho-alveolar lavage fluid, which indicated a significant role of fluid clearance on clinical outcome of patients with acute respiratory distress syndrome (ARDS) [28]. However, rates of water flux cannot be quantified with this method. This approach is not applicable to HAPE because of increased filtration of protein-rich fluid, whereas epithelia appear tight to protein in healthy individuals [12].

Based on these findings on the relation between active Na-reabsorption and pulmonary edema in rats it appears possible that alveolar Na^+^ reabsorption also plays a role in HAPE. Thus, a high basal activity of reabsorption would compensate excessive fluid accumulation caused by enhanced filtration, whereas insufficient reabsorption activity might promote alveolar edema [29].

Indirect evidence suggests that HAPE-susceptibles might have decreased Na-reabsorption in the lung. However, measurements were not possible at the alveolar epithelium, but at the nasal mucosa. This technique has been used to diagnose defective Cl-transport in cystic fibrosis [30,31], where the activity of Cl-secretion is decreased due to genetic variations of a Cl-channel called cystic fibrosis transmembrane regulator (CFTR) [32]. Use of this method on HAPE-susceptible subjects showed that the potential difference across the nasal mucosa (NPD) was lower in HAPE-susceptibles than in non-susceptibles in normoxia at low altitude [33]. Later, we confirmed this result [34] and also showed that the portion of the NPD caused by active Na-transport was lower, and that nasal epithelial Na-transport decreased even further at high-altitude (4559 m). If in fact these results indicate decreased Na-reabsorption in the alveoli, then exposure to hypoxia might start a vicious cycle in HAPE-susceptibles: hypoxic vasoconstriction increases fluid filtration into the alveolar space and intrinsically low Na-reabsorption in combination with hypoxic inhibition of transepithelial ion transport causes alveolar fluid accumulation. This further impairs oxygen diffusion and aggravates hypoxic pulmonary vasoconstriction. In sum, these events will eventually end in full-blown HAPE.

Indirect evidence for a protective role of an increased activity of Na-transport comes from treatment studies, where the inhaled β-adrenergic agonist salmeterol [33] and oral dexamethasone [2], both strong stimulators of alveolar epithelial reabsorption [21,35,36], significantly reduced the incidence of HAPE in HAPE-susceptible individuals. However, neither of these studies presented evidence for treatment-induced improvement of Na- and water-reabsorption. Furthermore, arguments for beneficial effects of improved alveolar reabsorption on pulmonary vascular tone in hypoxia are contradictory. Sartori et al. [33] found a pronounced improvement of arterial SO_2_ and PO_2_ in HAPE-susceptibles who received inhaled salmeterol during ascent and at high-altitude (4559 m) in comparison with a placebo group. Importantly, the treatment significantly reduced the incidence of HAPE in susceptible individuals. Yet, exaggerated sPAP did not decrease. For this reason the authors claimed that impaired alveolar Na-transport may cause HAPE-susceptibility, whereas improved Na-transport protects from HAPE [33]. However, this hypothesis has to be put into question. Less-severe hypoxic pulmonary arterial vasoconstriction and thus lower sPAP would be expected when stimulated reabsorption improves oxygenation. In fact, the authors of this study themselves published decreased sPAP in HAPE-susceptibles at high-altitude with salmeterol-inhalation in an abstract [37]. The discrepancy between results in the abstract [37] and their paper [33] has never been resolved. By contrast, oral dexamethasone [2] improved oxygenation and significantly decreased sPAP in HAPE-susceptibles. However, the action of this drug is not specific enough to conclude that stimulated alveolar reabsorption was the sole reason for improvement.

Along these lines, an interesting experiment has been performed on rats [38], where an aerosol containing the ENaC-inhibitor amiloride was administered to normoxic anesthetized rats at a dose that decreased alveolar fluid reabsorption by approximately 50%. This resulted in a significant increase in lung water within the first hour, decreased femoral arterial PO_2_, and increased right ventricular pressure. This finding is in accordance with the above mentioned hypothesis. However, although showing increased lung water and elevated right ventricular pressure also in hypoxic rats, this study did not demonstrate additivity of amiloride and hypoxia, a situation supposed to mimic the situation of HAPE-susceptibles, that is, intrinsically low Na-transport and hypoxic inhibition of reabsorption. Lack of additivity weakens the argument of a role of impaired Na-reabsorption as a cause of exaggerated hypoxic pulmonary vasoconstriction.

### 2.3. Oxygen Saturation and Pulmonary Vasoconstriction

If lung water and subsequent impairment of oxygen diffusion for whatever reason were the only mechanism accounting for exaggerated pulmonary arterial vasoconstriction then the increase in sPAP in HAPE might be estimated from the relation between SO_2_ and sPAP in healthy individuals determined by Luks et al. [39]. In this study arterial SO_2_ was varied by having subjects breathe hyperoxic and hypoxic gas mixtures both at sea level and at different locations at high-altitude. Their results indicate that the mean tricuspid-valve pressure gradient (TVPG) was higher at a given SO_2_ at the Everest Base Camp than at sea level, which may indicate some vascular remodeling during the slow ascent to high-altitude. However, the slope of the relationship between SO_2_ and TVPG was the same at either location indicating that the responsiveness to oxygen had not changed. Their results showed an increase in sPAP of approximately 6 mmHg per decrease in SO_2_ of 10% (Figure 2 in [39]). Applying this relation to studies performed in the Capanna Margherita (4559 m), a decrease in SO_2_ by 15%, which is quite normal at this altitude, would cause an increase in sPAP by approximately 9 mmHg, which is slightly less than measured values. The additional decrease in SO_2_ of approximately 10% found in HAPE would therefore account for an additional increase in sPAP of approximately 6 mmHg. However, sPAP values measured in HAPE are much higher, reaching values up to 100 mmHg. Furthermore, there appears to be no clear relation between sPAP and SO_2_ in HAPE (Figure 2).

Also results from short-term exposure to normobaric hypoxia point to altered vaso-reactivity in HAPE-susceptibles independent of differences in SO_2_. Dehnert et al. [17] have shown that sPAP was significantly higher in HAPE-susceptibles than in controls as early as 10 min after exposure to hypoxia, and that this gap widens with prolonged hypoxic exposure. At this early time-point there was no difference in SO_2_ (Figure 3) between these two groups indicating no diffusion limitation for oxygen and thus no accumulation of significant amounts of edema-fluid in the alveoli. Together these results point to some mechanism intrinsic to vascular smooth muscle cells exaggerating the pressure response in HAPE-susceptibles and that the diffusion limitation for oxygen is the consequence but not the initial trigger for HAPE. Mechanisms are not fully understood. It is unclear, to which extent an inhomogeneity of pulmonary vasoconstriction contributes to this setting [11,42]. Prevention of HAPE with tadalafil [2] and nifedipine [13,43] indicate altered NO-, cGMP-, and Ca-signaling. Interestingly, HAPE-susceptibles show impaired reactivity of peripheral arteries to the vasodilator acetylcholine in hypoxia, but not in normoxia, which might point to endothelial dysfunction and supports this notion [44], which might point to altered function of voltage-sensitive potassium channels. Lung vascular reactivity has not been studied using a similar approach.

## 3. Acute Mountain Sickness (AMS)

AMS may occur within the first 5 to 12 h after ascent to altitudes >2500 m with a prevalence of up to 25% below 3000 m. Prevalence is much higher at higher altitudes (e.g., ~50–85% at altitudes between 4500 and 5500 m) [6]. As with HAPE, slow ascent and pre-acclimatization decrease the risks, whereas individual susceptibility, that is, AMS in previous sojourns at high-altitude, increases the risk of developing AMS [45].

Symptoms are subjective. They are best evaluated with scoring systems like the Lake Louise score and the Environmental Symptoms Questionnaire (ESQ; AMS-C) [46,47]. The leading symptom of AMS is headache [48]. Accompanying symptoms are light-headedness, dizziness, loss of appetite, nausea, vomiting, fatigue, lassitude, trouble sleeping, and peripheral edema [49]. Successful prevention with acetazolamide and dexamethasone [6] allows no clear conclusion on the pathophysiology.

Many studies report decreased SO_2_ in hypoxia in individuals suffering from AMS in comparison with healthy controls, some of which suggest that decreased SO_2_ might be predictive of AMS (e.g., [1,50,51,52,53,54]). However, even in healthy individuals there is great variability in SO_2_ at high-altitude, which challenges this argument. Nevertheless, this finding raises the questions whether diffusion limitation for oxygen and the resulting aggravated hypoxemia are causal for the disease, and which mechanisms explain impaired oxygen diffusion.

### 3.1. AMS and Ventilation

As mentioned above, the degree of stimulation of ventilation is often measured as an indicator of the sensitivity of ventilatory control in hypoxia. Performing such tests on AMS-susceptible individuals provided varying results: no difference in the acute hypoxic ventilatory response (HVR) between controls and individuals who later developed AMS at 4559 m was found when exposure to hypoxia during the testing procedure was very short [15,55]. Extending exposure to the different levels of oxygen to 20–30 min, that is when hypoxic ventilatory depression can be expected to occur [14], a significantly lower SO_2_ was found in AMS-susceptible individuals in comparison to controls [50], which might indicate enhanced ventilatory depression in AMS. Also on days one to three at the Capanna Margherita (4559 m), Bärtsch et al. [55] found that poikilocapnic HVR did not increase in individuals who developed AMS, which one would have expected from the time course of ventilatory acclimatization and what had actually been found in healthy controls in this study [55]. Interestingly, end-tidal PCO_2_ was no different between controls and AMS despite the decreased HVR. Importantly, alveolar PO_2_ calculated from published data (Table 3 in [55]) using the alveolar gas equation was also no different, indicating that alveolar PO_2_ was within the normal range for this specific altitude. These results indicate normal (for the altitude) alveolar ventilation in AMS despite differences in HVR, which indicates that the diffusion limitation for oxygen cannot be caused by decreased alveolar PO_2_.

### 3.2. Lung Water Content and AMS

Bärtsch et al. [55] also report a higher alveolar-to-arterial oxygen difference (AaDO_2_) in those with severe AMS. Because of normal (for the altitude) alveolar PO_2_ this result indicates a diffusion limitation for oxygen resulting in decreased SO_2_ in AMS, which might be due to interstitial and/or alveolar edema. Extravascular lung water estimated from radiological and clinical tests and by measuring the closing volume was found to be higher in individuals with increased AMS-scores but without signs of HAPE at the Capanna Margherita (4559 m) [56]. However, in this study no difference in SO_2_ was found between individuals with and without indications of subclinical edema [56]. In a study performed at 3810 m, a small but significant association between AMS and lung water assessed with the B-line score from ultrasound measurements was reported [57]. However, also in this study there was no significant correlation between the B-line score and the change in SO_2_. This is surprising because a small increase in lung water is already thought to be sufficient to impair oxygen diffusion. This is in contradiction with the direct relation between the Lake Louise score and the AaDO_2_ mentioned above [55].

The reason for increased lung water in AMS is not clear. Increased filtration of fluid due to elevated capillary pressure and increased permeability, impaired lymphatic drainage, and impaired alveolar clearance might all contribute. Results on most of those mechanisms are scarce. Indirect evidence for increased filtration can be derived from a significant correlation between decreased SO_2_ in acute hypoxia and at high-altitude and increased sPAP in healthy individuals [39]. Based on an approximately 10% lower SO_2_ in AMS on day 2 at the Capanna Margherita (4559 m) [55], one might expect an elevation in sPAP in AMS too. Plotting (Figure 4) the results from the placebo group from a study performed at the Capanna Margherita (4559 m) showing that inhaled budesonide did not prevent AMS [41] indicates that the dependency of sPAP on SO_2_ was the same in healthy controls and in subjects with AMS and that this relation was very similar to the one reported by Luks et al. for healthy individuals [39]. In this study, SO_2_ was approximately 8% lower in AMS (*p* < 0.001), while sPAP was only 3 mmHg higher in AMS than in healthy individuals (*p* = 0.117). In a study performed at 3450 m in the Jungfraujoch Research Station on a possible protective effect of remote ischemic preconditioning [58], there was no significant difference in SO_2_ (0.386) and in sPAP (*p* = 0.934) between healthy subjects and those with AMS from the placebo group on the first morning after ascent. While on day three at high-altitude the difference in SO_2_ between controls and AMS was increased slightly (+3%; *p* = 0.041), there was no difference in sPAP between both groups. This indicates that despite clear symptoms of AMS (Lake Louise score >4 and AMS-C >0.7) there was only a mild impairment of oxygen diffusion at this altitude. The degree of hypoxia at 3450 m might have been too small to allow the detection of a difference in sPAP between controls and AMS with typical clinical approaches. Another reason might be the mode of ascent: while subjects ascended to the Capanna Margherita (4559 m) on foot, ascent to the Jungfraujoch Research Station (3450 m) occurred by train and lasted only 2 h, which raises the question of a possible impact of physical exercise on the degree of AMS and other physiological and clinical parameters.

It is unclear whether increased lung water indicated by decreased SO_2_ in some studies is a requirement for the occurrence of AMS. Looking at earlier time points is necessary to answer this question, which is only possible by comparison of results from high-altitude with chamber studies performed at equivalent degrees of hypoxia. Figure 5 summarizes the results from the placebo group of a study on short-term effects of ischemic preconditioning in normobaric hypoxia [59] and shows the time course of changes in the Lake Louise score for AMS and the respective change in SO_2_, and compares them with results from the placebo group from a study on the action of inhaled budesonide performed at an altitude of 4559 m [41]. It shows that symptoms of AMS become apparent already after 5 h in normobaric hypoxia and become stronger with prolonged stay. Also, at high-altitude symptoms were evident throughout the sojourn. However, in these early time-points in normobaric hypoxia there is no difference in SO_2_ between AMS and controls. Lower SO_2_ in subjects with AMS compared to controls seems to develop slowly after 18 h in normobaric hypoxia but is then present throughout the stay at high-altitude. Therefore, if SO_2_ in fact indicates a diffusion limitation for oxygen, likely by formation of subclinical edema, then these results indicate that AMS-symptoms appear before this diffusion limitation develops. Thus, aggravated decrease in SO_2_ appears not to be the primary trigger for AMS but rather a consequence.

## 4. Conclusions

Studying high-altitude-related illness in mountaineering settings includes gradually increasing severity of hypoxia and exertion during ascent. Often, complex measurements are not possible during ascent. In contrast, studies in normobaric hypoxia allow instant exposure to a constant level of decreased inspiratory oxygen, but in most cases physical exertion is lacking. Thus, results on physiological measures and on the development of high-altitude and hypoxia-related illness may be different between these two settings. Yet, combining them is the only way to assess the early phase of development of the illness and compare with the situation at high-altitude. This represents a major limitation of the analysis presented here.

Combination of these two settings indicates that SO_2_, which is a surrogate measure of oxygen diffusion across the alveolar wall when alveolar PO_2_ is normal (for the respective altitude), decreases slower than the illness develops; symptoms of AMS may appear hours before SO_2_ drops to levels that are normal for the altitude. Also, the exaggerated increase in sPAP, likely the main trigger causing HAPE, well precedes the decrease in SO_2_. Likely in both cases decreased SO_2_ indicative of (sub)-clinical edema develops after the initial trigger. This indicates that enhanced filtration from the pulmonary capillaries, impaired removal of lung water by impaired lymphatic drainage and by inhibited alveolar Na- and water reabsorption are likely not the cause of edema development. However, a pre-existing defect in one of those systems might act as an amplifier by increasing the degree of hypoxemia. Whereas it is quite obvious that increased lung water in HAPE is caused by enhanced filtration due to individual susceptibility to exaggerated hypoxic pulmonary arterial vasoconstriction, the reason for subclinical edema in AMS is less clear, and multiple pathophysiologic mechanisms have to be assumed. For both diseases, AMS and HAPE, those amplifying mechanisms require clarification because the resultant aggravation of hypoxemia might advance the illness.

## Figures and Tables

**Figure 1 ijms-20-04105-f001:**
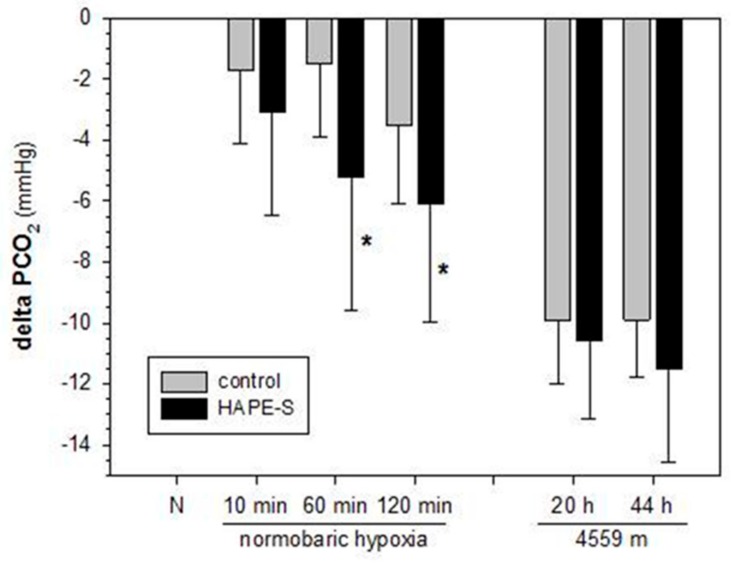
Time-course of decrease in partial pressure of carbon dioxide (PCO_2_) in controls and high-altitude pulmonary edema –susceptibles (HAPE-S) in hypoxia. The decrease in PCO_2_ was calculated as the difference between normoxia and the time-points in hypoxia for the respective studies. Data for normobaric hypoxia (equivalent to 4550 m) are so far unpublished end-tidal PCO_2_ values from 15 controls and 17 HAPE-S from a study on the time course of change in systolic pulmonary arterial pressure reported in acute hypoxia reported in [17]). Data on high-altitude at the Capanna Margherita (4559 m) represent the arterial PCO_2_ on day 2 of the sojourn from 10 controls and 9 HAPE-S from a study reported in [2]. Mean values ± standard deviation. N normoxia; * indicate significant difference between controls and HAPE-susceptibles.

**Figure 2 ijms-20-04105-f002:**
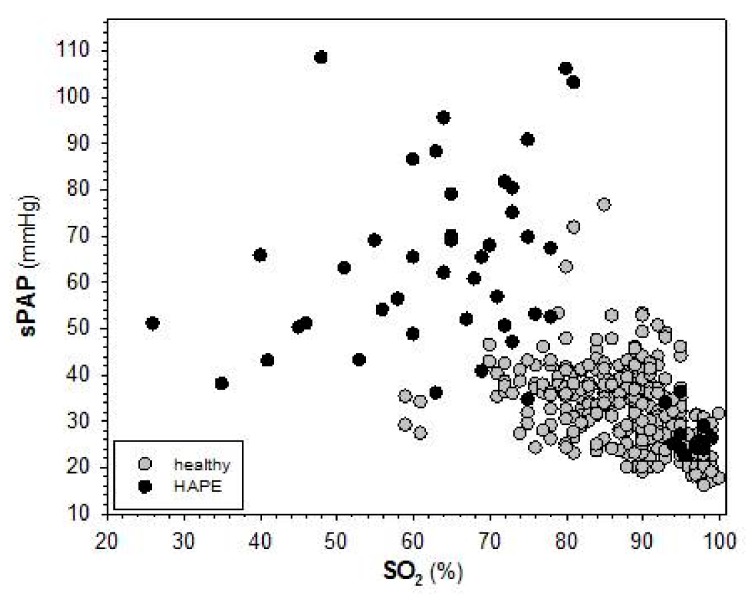
Lack of correlation between systolic pulmonary arterial pressure (sPAP) and SO_2_ in HAPE. Individual values from published studies [2,12,40,41], which had all been performed at the Capanna Margherita (4559 m), where subjects ascended by cable car to an altitude of 3200 m and climbed then to the Capanna Gnifetti (3600 m) in the afternoon and spent the night at this altitude. On the next morning they climbed to the Capanna Margherita (4559 m), where they spent two nights.

**Figure 3 ijms-20-04105-f003:**
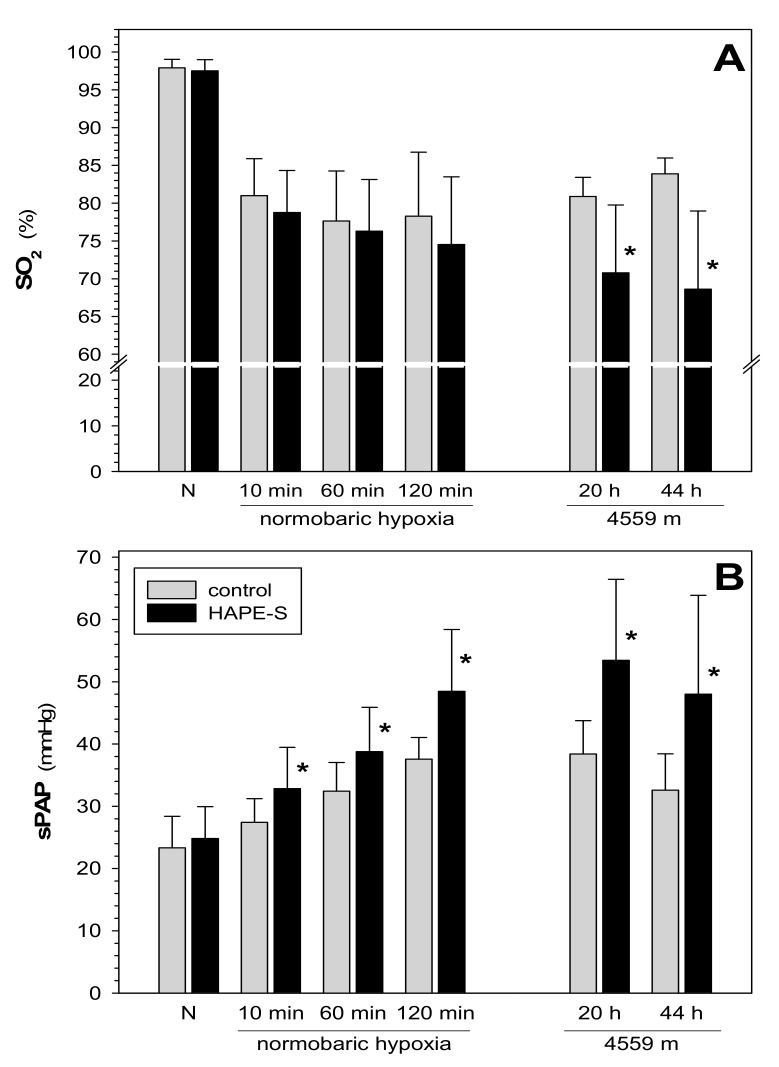
Time course of change in SO_2_ (**A**) and sPAP (**B**) in controls and HAPE-susceptibles (HAPE-S) during normobaric hypoxia (0–2 h) and at the Capanna Margherita (4559 m). Data on short-term normobaric hypoxia are from 15 controls and 17 HAPE-S from the study reported in [17], those from exposure to high-altitude (4559 m) are from 10 controls and 9 HAPE-S from reference [2]. In both studies, systolic pulmonary arterial pressure (sPAP) has been measured by Doppler echo-cardiography. Oxygen saturation (SO_2_) was measured by pulse-oximetry. Mean values ± standard deviation; N: normoxia; * indicates difference between controls and HAPE.

**Figure 4 ijms-20-04105-f004:**
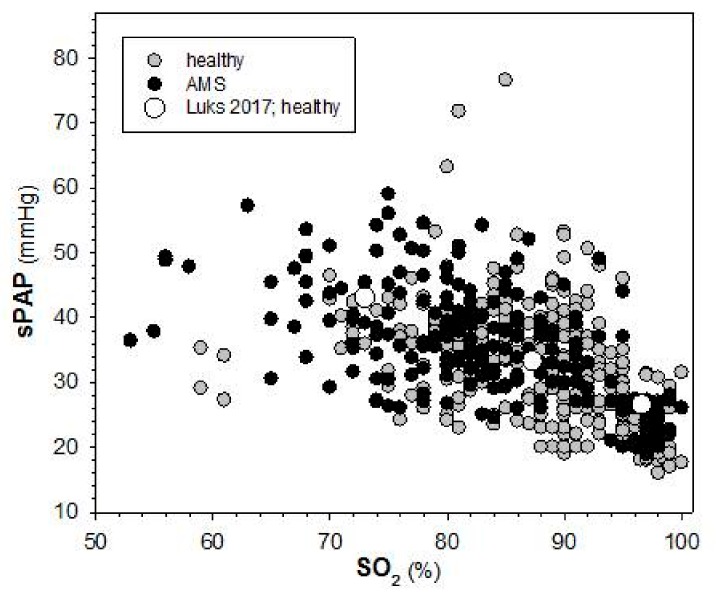
Relation between sPAP and SO_2_ in acute mountain sickness (AMS) and in healthy controls. Individual values from the placebo-group of a study on effects of ischemic preconditioning performed at 3450 m at the Jungfraujoch Research Station [58] and from the placebo group of a study on effects of inhaled budesonide at the Capanna Margherita (4559 m) [41]. Data split by AMS have not been reported in these publications. Ascent to 3450 m was by train, to 4559 m by cable car and climbing (see legend to Figure 1). Open circles are the mean values reported in [39], for comparison.

**Figure 5 ijms-20-04105-f005:**
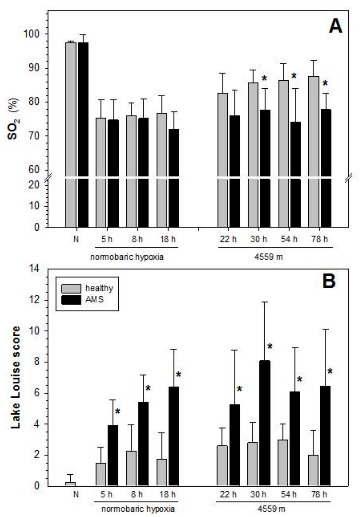
Time course of change in SO_2_ (**A**) and in the Lake Louise score (**B**) in controls and individuals developing AMS during normobaric hypoxia and at the Capanna Margherita (4559 m). Data on normobaric hypoxia are from a study on effects of remote ischemic pre-conditioning from 6 controls and 8 subjects with AMS reported in [17,59,60], those from exposure to high-altitude (4559 m) are from 6 controls and 8 subjects with AMS as reported in [41]. Oxygen saturation (SO_2_) was measured by pulse-oximetry. Mean values ± SD; N: normoxia; * indicates difference between controls and AMS.

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
