# Peer review of "The Hen or the Egg: Impaired Alveolar Oxygen Diffusion and Acute High-altitude Illness?"

_ijms, 2019, doi:10.3390/ijms20174105_

Round 1
Reviewer 1 Report
General comment
Mairbäurl and co-workers submitted a review article summarizing the currently available scientific literature regarding the pathophysiology of high altitude pulmonary edema (HAPE) and acute mountain sickness (AMS). With this regards the authors provide a comprehensive overview of studies investigating the (patho)physiology of parameters related to hemodynamics, alveolar ventilation and gas-exchange of individuals susceptible for HAPE/ AMS in comparison to healthy individuals. In addition, the effects of high altitude and oxygen levels on fluid clearance e.g. by ENaC-driven Na- reabsorption were discussed taking also cell culture and animal studies into consideration. From my perspective this manuscript is well written and provides a very good discussion regarding current concepts in the pathophysiology of HAPE/ AMS. Hence, I have only some minor comments.
Minor comments
Page 2, line 74: In HAPE, hypoxic pulmonary vasoconstriction is suggested to result in increased capillary pressure. From my understanding of the pathophysiology of pulmonary hypertension I would expect that hypoxic pulmonary vasoconstriction takes place in the pre-capillary arteries while an increase in the wedge pressure (capillary pressure) would require a post-capillary pathophysiology such as in veno-occlusive disease or left heart failure. Could you please comment on that and clarify in your review article?
Page 3, line 97-98: Similar to comment 1. “ …. Pulmonary arterial smooth muscle cells and subsequent exaggerated hypoxic vasoconstriction …. “. The link to the increased capillary pressure is not clear. I would expect that the arterial vasoconstriction protects the pulmonary capillary network from high pressures.
Figures 1, 3 and 5: Could you please add on how many individuals these data are based.
The subheadings in Section 2.1. and Section 2.2. are formatted differently – this should be harmonized.
Author Response
Reviewer #1:
Thank you for your helpful comments.
P2, l74: For clarification the sentence has been changed to “X-ray images of the thorax show visible infiltrates [7]. Doppler-echocardiographic [8] and right-heart catheter [9, 10] evaluations indicate exaggerated hypoxic pulmonary vasoconstriction (HPV) and increased capillary pressure [10] caused by veno-constriction or uneven distribution of blood flow and pre-capillary vasoconstriction resulting in capillary overperfusion in non-constricted areas [10, 11]. “
P3,l97-99: Please also see our response to the previous comment. To our opinion this aspect is different from the previous one because a lower PO2 at the vascular smooth muscle cells due to impaired oxygen diffusion should also result in stronger HPV. Capillary pressure is not mentioned here. We tried for better emphasis of this aspect. The sentence reads now like this: “This would result in an even further decreased PO2 at pulmonary arterial smooth muscle cells and subsequently exaggerated hypoxic vasoconstriction, which is a hallmark of HAPE-susceptibility [13].”
Numbers of subjects have been added to figure legends.
Formatting of subheadings has been adjusted.
Reviewer 2 Report
This review is interesting and deserves publication. Authors bring together interesting published data that they generally report and analyse with objectivity. However, they should emphasize that a low SO2 should not be used as a predictor to the development or as a sign of AMS and that a “good” arterial oxygen saturation can produce a false sense of security.
Most important comments:
- 21 : I do not agree to say that the decrease in SO2 is “pronounced” in those who develop AMS. The authors show that a decrease in SO2 is correlated with AMS, but in fact, this is a trend. It is clear that in some people, AMS symptoms develop while SO2 becomes lower than for teammates. However, on the terrain of altitude, this reviewer have seen many times that some individuals felt very well, performed normally, successfully climbed to the top of a mountain and never developed AMS with an SO2 lower than the mean of the group of climbers (sometimes clearly lower) and that some well saturated people suddenly develop AMS. On the other hand, a clear drop in SO2 is always present in cases of HAPE.
- 217: The potential role of O2-sensitive potassium current in PAP could be mentioned.
- Figure 3: bar colors are confusing between A and B. This should be changed or a specific legend should be included in Panel A.
- 247: “that decreased SO2 might be predictive of AMS” is really not justified and lead to unfair decisions.
- Figure 4: authors should comment on the signification of the results showing that “healthy” climbers present very low SaO2 or very high sPAP. A “true” climber (or patient) differs from a “mean” climber (or patient), which is a theoretical notion.
Other comments:
- 6: M. Berger (space)
- 28: “ before SO2 decreases ” is not correctly written as AMS and HAPE occur in hypoxia.
- 36: AMS and HAPE are pathologies with different symptoms.
- 72: in the development of HAPE, frothy sputum in usually white and becomes pink
- 191: SO2 was (space)
- 237: close the parenthesis
- 243: note that trouble sleeping is no more considered as a symptom of AMS.
- 313: only possible
- 344: I am not sure that “diffusion distance” is wright as the diffusion barrier is not only a question of distance
- 355: lack a period
- References
: Bärtsch is
sometimes with a umlaut and sometimes not.
Author Response
Reviewer #2.
Most important comments:
21: There is no doubt on decreased SO2 in individuals developing HAPE, and we certainly agree that there is considerable variability is SO2 in AMS. However, there are several studies showing a lower mean SO2 in AMS than in controls in the Margherita Hut (see references to fig. 5 and the respective data extracted from these studies, which show a decrease by approximately 10%). A decrease in mean SO2 implies that there might be very well individuals with only minor or no change in SO2 but also others with an even more pronounced decrease. We agree that a large variability in SO2 tolerance exists and that an individual SO2 is no good predictor and/or indicator for AMS. We have modified our text accordingly to express our and your concern about this argument.
Of course none of those details can be addressed in the abstract.
217: We have added a sentence on K-channels, but a bit later in the text than suggested. “226-7: supports this notion [44] which might point to altered function of voltage-sensitive potassium channels.”
247 and later: The author agrees with the uncertainty of SO2 being a predictor of AMS raised by the reviewer because of great inter-individual variability in SO2 in healthy and even greater variability in AMS, which might lead to false-positive decisions, unfair or not. We have moderated the statement to account for possible discrepancies: “Many studies report decreased SO2 in hypoxia in individuals suffering from AMS in comparison with healthy controls some of which suggest that decreased SO2 might be predictive of AMS (e.g. [1, 50-54]). “
Other comments:
28: thank you for this aspect: the sentence has been changed to “The resultant analysis reveals that in both, AMS and HAPE, the main pathophysiologic mechanisms are activated before aggravated decrease in SO2 occurs indicating that impaired alveolar epithelial function …”
36: “are different pathologies”: This is exactly what we were saying. However, we expanded beyond this by mentioning that both share decreased SO2 as a common symptom.
72: we have adjusted this sentence according to the reviewer’s statement: “.. and white, and as the illness progresses, pink frothy sputum.”
344: We certainly agree on this aspect. Diffusion distance might be correct if (sub)-clinical edema were the only reason. We argued for the latter because of most likely normal alveolar PO2. Nevertheless, we have moderated this statement to: “Combination of these two settings indicates that SO2, which is a surrogate measure of oxygen diffusion across the alveolar wall when alveolar PO2 is normal… “
Thank you for pointing to the inconsistency in bar-colors in fig.3; has been adjusted.
Formatting errors and misspellings of Bärtsch have been corrected.